# Learning to Disprove: Formal Counterexample Generation with Large Language Models

## Abstract

Mathematical reasoning demands two critical, complementary skills: constructing rigorous proofs for true statements and discovering counterexamples that disprove false ones. However, current AI efforts in mathematics focus almost exclusively on proof construction, often neglecting the equally important task of finding counterexamples. In this paper, we address this gap by fine-tuning large language models (LLMs) to reason about and generate counterexamples. We formalize this task as formal counterexample generation, which requires LLMs not only to propose candidate counterexamples but also to produce formal proofs that can be automatically verified in the Lean 4 theorem prover. To enable effective learning, we introduce a symbolic mutation strategy that synthesizes diverse training data by systematically extracting theorems and discarding selected hypotheses, thereby producing diverse counterexample instances. Together with curated datasets, this strategy enables a multi-reward expert iteration framework that substantially enhances both the effectiveness and efficiency of training LLMs for counterexample generation and theorem proving. Experiments on three newly collected benchmarks validate the advantages of our approach, showing that the mutation strategy and training framework yield significant performance gains.

## 1 Introduction

Endowing machines with mathematical reasoning capabilities has long represented one of the most fundamental and challenging frontiers in artificial intelligence research (Ahn et al., 2024). In recent years, with the emergence of reasoning large language models (LLMs) such as OpenAI-o1 (OpenAI, 2023), DeepSeek-R1 (Guo et al., 2025), and Gemini-2.5-Pro (Comanici et al., 2025), significant advances have been achieved in this area. Furthermore, several formal reasoning LLMs, including Seed prover (Chen et al., 2025a), Kimina prover (Wang et al., 2025), and Goedel prover (Lin et al., 2025b), have also demonstrated strong performance in generating formal proofs and interacting with theorem provers (De Moura et al., 2015; Barras et al., 1999; Paulson et al., 1993). The impressive capabilities exhibited by these models open up new avenues for addressing advanced and complex mathematical problems. However, existing studies often emphasize the logical deduction of LLMs; their capability to find counterexamples remains underexplored (Li et al., 2025).

In mathematics, counterexamples are far more than mere exceptions (Barahmand, 2019); they play a vital role in theory development (Lander et al., 1967), conjecture refinement (Haselgrove, 1958), and educational enhancement (Zazkis & Chernoff, 2008). For instance, since a conjecture must hold for all possible inputs, identifying anomalous examples and special cases is a crucial step for formulating valuable conjectures. Moreover, for LLM reasoning, its effectiveness largely depends on how well models can reflect upon and evaluate their intermediate reasoning steps (Renze & Guven, 2024). Hence, counterexample generation becomes equally vital: it equips models with the ability to self-verify their logical processes, thereby strengthening both their reasoning reliability and explanatory power.

In this paper, we focus on training LLMs to generate formal counterexamples. Although casting the task into a formal setting requires the LLM to additionally provide a formal proof for each counterexample, it allows for the automatic evaluation of the proposed counterexample via the Lean 4 theorem prover. Two significant challenges arise in this task: First, there is a severe scarcity of training data. Currently, CounterMath (Li et al., 2025) stands as the sole dataset specifically curated

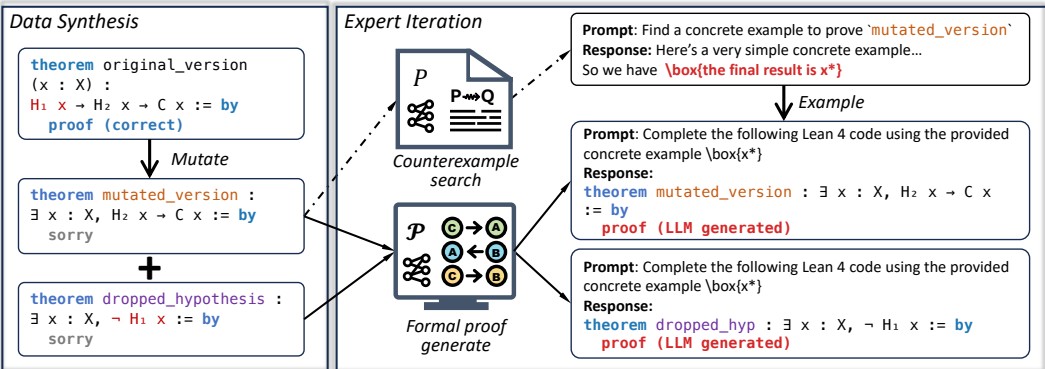

Figure 1: Framework of counterexample training. In the data synthesis stage, the symbolic mutation drops the hypothesis of a provable theorem, creating new counterexample problems. In the subsequent expert stage, two rewards are introduced based on whether the generated counterexample can prove the mutated version and dropped hypothesis, boosting training effectiveness and efficiency.

for benchmarking counterexample generation, comprising only 1,216 natural language problems. This limited size renders it inadequate for practical LLM training. Second, the reward signals during training are exceptionally sparse. When LLMs fail to produce correct counterexamples for complex problems, the training reward vanishes, hindering further improvement in model performance and resulting in a plateau at a low success rate.

To address these challenges, we propose an integrated framework illustrated in Figure 1. We design a new mutation strategy that discards the necessary hypothesis for any provable theorem, thereby invalidating it and inducing counterexamples. Through this mutation strategy, we synthesize a large number of counterexample problems, significantly enriching the dataset. Furthermore, we introduce a multi-reward function based on the proven success rate of both the extracted problem and discarded hypothesis, ensuring that the reward remains valid even when LLMs solve hard problems.

Experiments on three newly collected benchmarks demonstrate the effectiveness of our framework. Leveraging the proposed mutation strategy, we synthesized 575K counterexample data points for fine-tuning. Together with a curated large-scale dataset and a multi-reward training scheme, this allows us to train a model specialized for counterexample generation. We further establish three applicable tasks: counterexample search, verification of autoformalized results, and verification of reasoning steps. Experimental results show that our fine-tuned model achieves significant improvements on these tasks over current LLMs. Notably, our fine-tuned model achieves a relative improvement of 47% to 74% in pass@1 success rate compared to the strongest baseline.

## 2 FORMAL COUNTEREXAMPLE GENERATION

The counterexample problem arises in disproving a universal mathematical conjecture of the form $\forall x, P(x)$. By identifying a counterexample $x^*$, the conjecture can be invalidated through its negation $\exists x, \neg P(x)$, which can then be reframed as an instantiated version $P(x^*)$. A classic example is the conjecture "all continuous functions are differentiable." To refute this claim, one must demonstrate the existence of a function that is continuous but non-differentiable, e.g., the absolute value function $f(x) = |x|$. Recently, automated mathematical research and knowledge discovery have garnered increasing attention (Boiko et al., 2023; Romera-Paredes et al., 2024; Novikov et al., 2025). As a result, the counterexample generation, as a specific yet crucial step in judging the correctness and completeness of newly proposed statements, has also evolved from being merely an incremental improvement to becoming pivotal in shaping the future of mathematical inquiry.

There are several symbolic tools designed for finding counterexamples to conjectures, such as *nitpick* in Isabelle (Blanchette & Nipkow, 2010) and *plausible* in Lean 4 (Lean Prover Community, 2024). They often rely on sophisticated software engineering techniques like random testing or SAT/SMT solving (Bérard et al., 2013; Ammann & Offutt, 2017). Nonetheless, mathematics is fundamentally different from program proving, and the inherent complexity of higher-order logic hinders the

Figure 2: Task of formal counterexample generation. This task requires the LLM first to perform informal reasoning to identify a valid counterexample for the given problem, and then generate the corresponding formal proof, which is automatically verified by theorem provers (e.g., Lean 4).

effectiveness of these tools in both formulating problems and generating solutions. A promising alternative lies in LLM-based counterexample generation. LLMs offer substantial expressiveness and strong reasoning capabilities, albeit with some sacrifice in soundness compared to symbolic tools.

In addition to using LLM-based counterexample generation as a powerful tool for examining mathematical conjectures, training LLMs to generate counterexamples offers a new perspective for boosting the LLM's reasoning capabilities. Unlike traditional mathematical proofs that are typically derived through logical deduction, aligning well with existing chain-of-thought models (Wei et al., 2022; Chen et al., 2025b), counterexample generation involves a guess-and-check paradigm (Young, 2009; Redish, 2016); the LLM first proposes a potential counterexample and then verifies its validity.

Formal counterexample generation aims to establish a two-stage, informal-to-formal process. As illustrated in Figure 2, given a formal problem, the LLM is first prompted to find a counterexample through informal reasoning. It then produces a formal proof for the theorem based on this derived counterexample. Finally, the validity of the proposed counterexample is confirmed if its corresponding formal proof is successfully verified by the Lean 4 theorem prover (De Moura et al., 2015).

This approach differs from other informal-to-formal reasoning methods, which mainly focus on creating an informal sketch followed by its formalization and completion (Jiang et al., 2023; Ren et al., 2025; Cao et al., 2025). In those methods, informal and formal reasoning are consistent and congruous with one another. In contrast, formal counterexample generation necessitates a sequential pattern of informal-to-formal reasoning. Specifically, the initial natural language reasoning serves as a counterexample proposal, transforming the abstract challenge of proving an existential statement into the more concrete task of formally verifying a specific instance.

Nevertheless, both stages of formal counterexample generation pose distinct challenges for current LLMs. The "guess" phase requires models to propose effective counterexamples by analyzing various edge scenarios, including corner cases and limit values. This particular capability remains an undeveloped aspect of their training (Li et al., 2025). In the "check" phase, LLMs generate formal proofs, an endeavor complicated by the scarcity of training data for existential theorems (Ying et al., 2024), which significantly hinders the models' performance. By addressing these challenges, counterexample generation can become a powerful tool to enhance the reasoning capabilities of LLMs and assist mathematicians in verifying conjectures.

## 3 THE PROPOSED FRAMEWORK

Our proposed approach consists of two stages: (1) counterexample problem synthesis and (2) multi-reward guided training. The first stage is devoted to creating a large-sized unlabeled dataset, i.e., the counterexample problem without a ground-truth answer and a formal proof. In the second stage, we carry out the expert iteration, where the LLM repeatedly generates counterexamples and formal proofs and then is fine-tuned on the successful ones. We will outline each stage as follows.

### 3.1 COUNTEREXAMPLE PROBLEM SYNTHESIS

We first collect a substantial number of formal theorems to serve as foundational seeds. These seed theorems must meet two criteria: they should be (1) formally provable and (2) expressed in a universal format (e.g., $\forall x$). In addition to sourcing these from existing Lean 4 formal libraries,

like Mathlib, we extract intermediate subgoals or lemmas from formal proofs generated by LLMs. Specifically, given a formal proof, we decompose it into a series of proof steps, and categorize each proof step by its proving style. If a proof step is stated in declarative style (i.e., using *have* or *suffices* tactic), we directly transform it into a new theorem. Conversely, for a proof step stated in a procedural style (using tactics such as *simp* and *rw*), we analyze and compare the proof states before and after applying such a tactic to establish a new theorem.

Next, we design a mutation-based data synthesis method. To elaborate, let us start with a formally provable, universally stated theorem, which can be represented as

```
theorem original_version (x : X) : H₁ x → H₂ x → C x
```

Here, $H_1$ and $H_2$ are two hypotheses, and $C$ is the conclusion. This theorem asserts that for any element $x$ in the space $X$, if both $H_1(x)$ and $H_2(x)$ hold true, then $C(x)$ follows.

From this theorem, we can construct a new theorem by discarding one of the hypotheses (for example, $H_1(x)$), resulting in $H_2(x) \to C(x)$. If $H_1(x)$ is indeed essential to the original theorem, the mutated version will be invalid, indicating the existence of a counterexample. We can thus formalize this counterexample problem as

```
theorem mutated_version : ∃ x : X, H₂ x → C x
```

To ensure the necessity of the omitted hypothesis, we propose initially using the Lean 4 theorem prover to analyze the formal proof of the original theorem (Lean Prover Community, 2025), identifying and eliminating any redundant hypotheses. Notably, compared to existing methods that merely prompt LLMs to propose or rephrase problems for data augmentation (Wang et al., 2024b; Yu et al., 2023; Huang et al., 2024), this rigorous approach guarantees the success of our mutations and the validity of the generated problems. By leveraging this mutation strategy, we can produce a substantial number of counterexample problems for subsequent LLM training.

## 3.2 MULTI-REWARD GUIDED TRAINING

We follow the standard expert iteration pipeline (Anthony et al., 2017) to train the LLMs with unlabeled data. In this framework, the model is repeatedly applied to perform large-scale inference on counterexample problems, from which candidate solutions and formal proofs are collected. The correct ones are then used to further refine the model through supervised fine-tuning. A key limitation, noted in prior work, is that LLMs often fail to produce correct results on difficult problems (Polu et al., 2022; Wu et al., 2024b; Dong & Ma, 2025), restricting training to a cycle of generating and fine-tuning only on simpler examples. This issue is essentially the sparse reward problem in classic reinforcement learning (Hare, 2019; Riedmiller et al., 2018), commonly alleviated through techniques such as reward shaping (Laud, 2004) or curriculum learning (Bengio et al., 2009).

Rather than adopting these conventional approaches, we propose using a multi-reward strategy to effectively address this limitation. Specifically, it can be observed that a valid counterexample for the mutated version $\exists x, H_2(x) \to C(x)$ should also contradict the removed hypothesis, i.e., $\neg(H_1(x))$ holds for the same counterexample. This leads to the formation of a new theorem

```
theorem dropped_hypothesis : ∃ x : X, ¬ H₁ x
```

To compute the score of the generated counterexample $x^*$, the LLM first provides the respective formal proofs for the two theorems *dropped_hypothesis* and *mutated_version*. These proofs are subsequently verified by the theorem prover, which yields two distinct rewards for the counterexample. The first reward is based on whether $x^*$ proves the necessary condition $\neg H_1(x^*)$, while the second reward is determined by whether it proves the target theorem $H_2(x^*) \to C(x^*)$. Since the necessary condition can be satisfied directly and its proof is straightforward to construct, the first reward is considerably easier to activate. This double-reward mechanism successfully guarantees the effect of the reward, even in instances where the counterexample cannot be conclusively determined. Ultimately, we combine these two rewards as the final sample weight for the supervised fine-tuning.

Additionally, we include the training of LLMs for generating formal proofs of counterexamples in the expert iteration. Besides collecting the formal proof of *mutated_version*, the formal proof of

*dropped_hypothesis* can also be used for fine-tuning. However, since formally proving the necessary condition is often easier, the model might fall into a low-difficulty trap, frequently generating shorter proofs. To mitigate this, the sample weight of the first formal proof should be reduced, and other techniques, e.g., group relative policy optimization (GRPO) (Shao et al., 2024), can also be employed to encourage the generation of longer formal proofs.

## 3.3 OVERALL FRAMEWORK

Formally, let us define the seed dataset as $\{\mathcal{T}_i\}_{i=1}^N$, where $\mathcal{T}_i$ stands for a seed theorem. For every theorem $\mathcal{T}_i$, we apply the mutation strategy and obtain the mutated version $\mathcal{M}_i$ and the dropped hypothesis $\mathcal{H}_i$. We introduce two LLMs, represented as $q_\phi$ and $q_\psi$, for generating counterexamples and formal proofs, respectively. For a given counterexample problem $\mathcal{M}_i$, we first prompt the LLM to propose a counterexample $x_i \sim q_\phi(x_i \mid \mathcal{M}_i)$. Next, the formal proof $\pi_i^{\mathcal{M}}$ is constructed by integrating the problem and the proposed counterexample, i.e., $\pi_i^{\mathcal{M}} \sim q_\psi(\pi_i^{\mathcal{M}} \mid \mathcal{M}_i, x_i)$. In addition, the LLM provides the formal proof $\pi_i^{\mathcal{H}}$ for the dropped hypothesis, given by $\pi_i^{\mathcal{H}} \sim q_\psi(\pi_i^{\mathcal{H}} \mid \mathcal{H}_i, x_i)$.

We utilize the indicator function $\mathbb{I}(\cdot)$ to signify the theorem prover's verification. Through verifying the two formal proofs $\pi_i^{\mathcal{M}}$ and $\pi_i^{\mathcal{H}}$, we establish the combined reward for the counterexample $x_i$ as $r_i := r_i^{\mathcal{M}} + r_i^{\mathcal{H}}$, where $r_i^{\mathcal{M}} := \alpha \mathbb{I}(\pi_i^{\mathcal{M}})$, $r_i^{\mathcal{H}} := (1-\alpha)\mathbb{I}(\pi_i^{\mathcal{H}})$, and $\alpha \in [0, 1]$ is a trade-off coefficient. Then, the supervised fine-tuning of counterexample generation is conducted on the weighted dataset $\{(\mathcal{M}_i, x_i; r_i)\}_{i=1}^N$. We continue to use these two rewards as the weights for the supervised fine-tuning of formal proof generation. The corresponding weighted dataset is derived from the formal proofs of two theorems and structured as $\{((\mathcal{M}_i, x_i), \pi_i^{\mathcal{M}}; r_i^{\mathcal{M}})\}_{i=1}^N \cup \{((\mathcal{M}_i, x_i), \pi_i^{\mathcal{H}}; r_i^{\mathcal{H}})\}_{i=1}^N$.

The overall framework is accordingly summarized in Algorithm 1.

---

**Algorithm 1** The Integrated Workflow of Data Mutation and Multi-Reward Training

---

**Input:** A seed dataset $\{\mathcal{T}_i\}_{i=1}^N$, with each formally provable and expressed in a universal format.
**Output:** Two LLMs, $q_\phi$ and $q_\psi$, for generating counterexamples and formal proofs.

1: *# Phase 1: Dataset Mutation*
2: **for** $i = 1, \ldots, N$ **do**
3:     Mutate $\mathcal{T}_i$, collecting the mutated version $\mathcal{M}_i$ and the dropped hypothesis $\mathcal{H}_i$.
4: **end for**

5: *# Phase 2: Expert Iteration*
6: **for** $k = 1, \ldots$ **do**
7:     *# Step 2.1: Large-scale Inference and Verification*
8:     **for** $i = 1, \ldots, N$ **do**
9:         Generate a counterexample $x_i$ using using $q_\phi$: $x_i \sim q_\phi(x_i \mid \mathcal{M}_i)$.
10:         Generate formal proofs for the mutated statement and the dropped hypothesis:
11:         $\pi_i^{\mathcal{M}} \sim q_\psi(\pi_i^{\mathcal{M}} \mid \mathcal{M}_i, x_i)$ and $\pi_i^{\mathcal{H}} \sim q_\psi(\pi_i^{\mathcal{H}} \mid \mathcal{H}_i, x_i)$.
12:         Compute rewards based on proof correctness:
13:         $r_i^{\mathcal{M}} := \alpha \cdot \mathbb{I}(\text{Verify}(\pi_i^{\mathcal{M}}))$, $r_i^{\mathcal{H}} := (1 - \alpha) \cdot \mathbb{I}(\text{Verify}(\pi_i^{\mathcal{H}}))$, and $r_i := r_i^{\mathcal{M}} + r_i^{\mathcal{H}}$.
14:     **end for**
15:     *# Step 2.2: Supervised Model Fine-tuning*
16:     Update the model $q_\phi$ based on the weighted dataset $\{(\mathcal{M}_i, x_i; r_i)\}_{i=1}^N$.
17:     Update the model $q_\psi$ on the dataset $\{((\mathcal{M}_i, x_i), \pi_i^{\mathcal{M}}; r_i^{\mathcal{M}})\}_{i=1}^N \cup \{((\mathcal{M}_i, x_i), \pi_i^{\mathcal{H}}; r_i^{\mathcal{H}})\}_{i=1}^N$.
18: **end for**

---

## 4 EXPERIMENTS

This section presents the experimental results. Specifically, our study aims to address the following research questions (RQs):

- **RQ1 (Efficacy and efficiency of data mutation):** What is the success rate and time consumption of data mutation for generating new theorems from a single seed theorem?

- **RQ2 (Efficacy and efficiency of multi-reward training):** Compared to single-reward training, does multi-reward training improve the performance and training efficiency?

| Source | # of seed theorems | # of mutated problems |
|---|---|---|
| Mathlib | 43,990 | 88,217 |
| Leanworkbook | 17,061 | 42,480 |
| MiniF2F | 48,415 | 94,215 |
| PutnamBench | 212,463 | 350,127 |
| Total | 321,929 | 575,039 |

Table 1: Overall results of data mutation. Applied to a set of seed theorems, our mutation method successfully generates a diverse collection of counterexample instances.

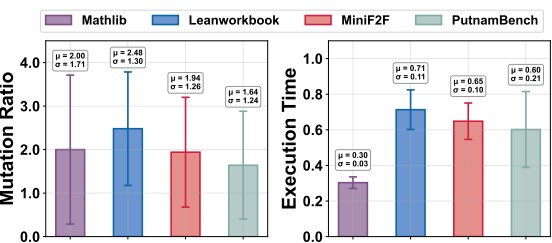

Figure 3: Results of mutation ratio and average execution time. Evaluated across multiple data sources, our method maintains a high mutation ratio while remaining computationally efficient.

- **RQ3 (Performance improvement of integrated workflow):** Compared with baselines, does our proposed framework achieve better performance in generating formal counterexamples?

All experiments were conducted on a Linux server featuring two AMD EPYC 7C13 64-core processors equipped with eight high-end GPUs. The code, together with the experimental data, is available at `https://figshare.com/s/02e05a2c2945ee10dcc4`.

## 4.1 RQ1: EFFICACY AND EFFICIENCY OF DATA MUTATION

Leveraging the proposed mutation strategy, we can curate a substantial number of counterexample problems as training materials for LLMs. To obtain the seed theorems, we aggregated all theorems and lemmas from the Lean 4 formal library Mathlib (mathlib Community, 2020) and the popular formal dataset Leanworkbook (Ying et al., 2024). We also extracted intermediate subgoals or lemmas from formal proofs generated by LLMs (Goedel prover v2 (Lin et al., 2025b) and DSP+ (Cao et al., 2025)) on the MiniF2F (Zheng et al., 2021) and PutnamBench (Tsoukalas et al., 2024). These four data sources originate from diverse mathematical domains and varying levels of difficulty. Mathlib encompasses a broad spectrum of mathematical concepts and foundational lemmas, while Leanworkbook comprises formalized content from approximately ten distinct categories (e.g., algebra, number theory, combinatorics). MiniF2F and PutnamBench focus on competition-level problems, with the former targeting high-school-level challenges and the latter undergraduate-level tasks, potentially involving more complex concepts.

We implemented a Lean 4 tactic, *mutate*, to perform the symbolic mutation. Before applying it, we utilized the Lean 4 theorem prover to analyze the formal proof of the seed theorem, identifying and eliminating any redundant hypotheses. Following the execution of the mutate operator, the theorem prover was invoked again to ensure the grammatical correctness of the newly generated problem.

The results of the mutation process on the four data sources are presented in Table 1. In total, we collected approximately 575K counterexample problems, which constitute a sufficiently large dataset to support LLM fine-tuning. The efficacy and efficiency of the mutation procedure are illustrated in Figure 3. The proposed method yields a mutation ratio in the range of 1.65 - 2.48, while the average execution time per seed theorem remains between 0.3 and 0.71 seconds. Note that the execution time on Mathlib is comparatively lower because this formal library is pre-compiled.

## 4.2 RQ2: EFFICACY AND EFFICIENCY OF MULTI-REWARD TRAINING

To evaluate the proposed multi-reward training, we conducted an ablation study by comparing training with and without the multi-reward function. To elaborate, we selected Qwen3 8B as the base model (Qwen Team, 2025) for informal reasoning, which is responsible for proposing counterexamples, and DeepSeek-Prover-v2 7B (Ren et al., 2025) as the base model for formal reasoning, which generates proofs for the proposed counterexamples. Both models were trained under the framework described in Algorithm 1, with the sole difference between the single-reward and multi-reward settings being the choice of coefficient $\alpha$. In the single-reward setting, $\alpha$ was fixed at 1.0, whereas in the multi-reward setting, it was set to 0.8 to ensure that both rewards contribute effectively. We also

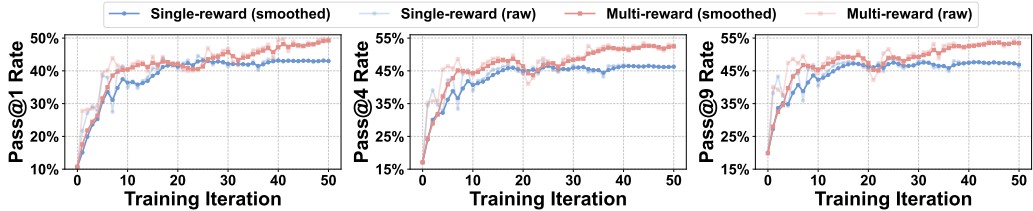

Figure 4: Pass@$k$ ($k = 1, 4, 9$) rate curves on the validation set. Compared with the single-reward baseline, multi-reward training converges faster and yields superior final performance.

adapted Dr.GRPO (Liu et al., 2025), a normalized variant of GRPO, to regulate proof length, thereby preventing proofs from being excessively short (overfitting) or unnecessarily long (overthinking).

From the collected dataset of 575K instances, we reserved 3K for validation. Due to limited computational resources, we trained for only a single epoch. In this epoch, we employed 56 iterations with a batch size of 10K, ensuring that each instance was utilized exactly once. In each iteration, large-scale inference and verification were conducted, after which both models were updated accordingly.

We compute pass@$k$ rates for $k = 1, 4, 9$ on the validation set and plot the pass@$k$ rate curves in Figure 4. Compared with the single-reward training, the multi-reward approach achieves both faster convergence and higher final performance. Specifically, at convergence, multi-reward training reaches approximately 49%, 52%, and 54% for pass@1, pass@4, and pass@9, respectively, whereas the single-reward baseline attains only 43%, 46%, and 47%. For reference, we also report the pass@$k$ results of existing LLM-based theorem provers in Appendix D.

### 4.3 RQ3: PERFORMANCE IMPROVEMENT OF INTEGRATED WORKFLOW

We establish three tasks to evaluate our fine-tuned model: the identification of counterexamples, as well as the verification of autoformalized results and reasoning steps. For the counterexample identification, we employ Kimina-Autoformalizer 7B (Wang et al., 2025) to formalize CounterMath (Li et al., 2025), which is constructed from four classic counterexample textbooks. This yields 1,058 formal counterexample problems, forming our benchmark (denoted as FOR-COUNTER). Furthermore, we assess the effectiveness of counterexample generation by identifying potential errors in the formalization of correct theorems and in the reasoning steps of proving theorems. Therefore, we construct VERI-FORMALIZE, which comprises 3K unprovable problems generated by formalizing FormalMath (Yu et al., 2025), and another 3K dataset VERI-FORMALIZE obtained when applying DSP+ (Cao et al., 2025) to solve the problems from FormalMath.

The pass@$k$ ($k = 1, 4, 9$) results are summarized in Table 2. Instead of reporting pass@$k$ rates, we present the absolute number of correctly solved problems, since the established benchmark contains inaccuracies. The results show that the model fine-tuned with our framework significantly outperforms existing state-of-the-art LLMs. Overall, open-source neural provers demonstrate stronger performance than proprietary, general-purpose reasoning models. Our fine-tuned model further advances the state of the art: for pass@1, it solves 95, 69, and 63 more problems than the strongest baseline across the three benchmarks. Similar gains are observed for pass@4 and pass@9, further confirming the superiority of our model. We also conduct an ablation study in Appendix D to evaluate the contribution of each component (i.e., informal and formal reasoning models).

## 5 RELATED WORK

Formal counterexample generation is generally related to LLM-based counterexample generation and formal reasoning (a.k.a neural theorem proving). We provide a brief overview of related work in these directions and highlight the connections and differences between our work and theirs.

### 5.1 LLM-BASED COUNTEREXAMPLE GENERATION

Counterexample finding is essential for both advancing mathematical research and strengthening the reasoning capabilities of LLMs. Despite its importance, there has been relatively little work on in-

Table 2: Pass@$k$ ($k = 1, 4, 9$) of counterexample generation on three tasks. Our fine-tuned model achieves the best performance on all three benchmarks and all three $k$ values, outperforming both proprietary reasoning models and open-source neural theorem provers.

| Dataset | FOR-COUNTER | | | VERI-REASON | | | VERI-FORMALIZE | | |
|---|---|---|---|---|---|---|---|---|---|
| | ✓@1 | ✓@4 | ✓@9 | ✓@1 | ✓@4 | ✓@9 | ✓@1 | ✓@4 | ✓@9 |
| PROPRIETARY REASONING MODELS | | | | | | | | | |
| Gemini-2.5-Flash | 21 | 66 | 82 | 5 | 9 | 15 | 2 | 8 | 13 |
| Grok-3-mini | 30 | 74 | 101 | 4 | 10 | 19 | 2 | 8 | 19 |
| GPT-4.1-mini | 19 | 65 | 103 | 54 | 97 | 150 | 31 | 87 | 137 |
| Deepseek-R1 | 61 | 135 | 158 | 51 | 75 | 105 | 27 | 72 | 102 |
| OPEN-SOURCED NEURAL PROVERS | | | | | | | | | |
| Leanabell-prover | 106 | 159 | 181 | 144 | 210 | 231 | 111 | 198 | 228 |
| STP-prover | 101 | 157 | 179 | 131 | 151 | 170 | 99 | 150 | 171 |
| Kimina-prover-distill | 31 | 109 | 165 | 66 | 114 | 156 | 18 | 141 | 249 |
| Goedel-prover-v2 | 89 | 177 | 215 | 88 | 147 | 200 | 63 | 165 | 201 |
| Deepseek-prover-v2 | 127 | 200 | 224 | 144 | 203 | 234 | 69 | 135 | 186 |
| Ours | **222** | **274** | **302** | **213** | **260** | **295** | **174** | **255** | **313** |
| Δ | 95 74%↑ | 74 37%↑ | 78 34%↑ | 69 47%↑ | 50 23%↑ | 61 26%↑ | 63 56%↑ | 57 28%↑ | 64 25%↑ |

corporating counterexample generation and AI techniques. Early attempts (Wagner, 2021; Ghebleh et al., 2024a;b; Roucairol & Cazenave, 2022; Vito & Stefanus, 2023) employed reinforcement learning and Monte Carlo tree search to identify counterexamples in combinatorics and graph theory, later extending this line of work with transformer-based models to improve search performance (Charton et al., 2024). More recently, some studies have begun to explore the potential of LLMs in proposing counterexamples. For example, CounterMath (Li et al., 2025) introduced a benchmark in this direction, demonstrating that current LLMs still face significant challenges. However, this benchmark is established purely in natural language, making it difficult to rigorously verify the correctness of counterexample candidates. MathConstruct (Balunović et al., 2025), by contrast, evaluates constructive proof generation and integrates symbolic methods for validation. Compared with our work, it is confined to competition-level problems, whereas our work grounds the task in Lean 4 formal language and focuses on undergraduate-level settings.

## 5.2 LLM-BASED FORMAL REASONING

The mathematical reasoning capability of LLMs is widely regarded as a key indicator of their overall intelligence (Ahn et al., 2024; Asperti et al., 2025; Pantsar, 2025). Formal reasoning requires an LLM to express its thought process in a formal language, with theorem provers verifying each step (Yang et al., 2024; Li et al., 2024). Leveraging advanced techniques such as autoformalization (Wu et al., 2022), premise retrieval (Yang et al., 2023), library learning (Wang et al., 2023), and proof search (Lample et al., 2022), LLM-based formal reasoning has achieved remarkable progress (Ji et al., 2025; Cao et al., 2025; Ren et al., 2025; Lin et al., 2025b), culminating in silver-medal level performance at the 2025 International Mathematical Olympiad (Chen et al., 2025a).

However, to the best of our knowledge, systematic investigations into the formal proof of counterexample problems remain extremely limited. Although some methods involve disproving the newly collected theorem when it cannot be successfully proved (Jiang et al., 2024; Lin et al., 2025b; Ren et al., 2025), the majority of available data do not fall into this category. As a result, existing LLM-based theorem provers perform poorly on counterexample-guided formal reasoning. Our work explicitly targets this gap by synthesizing counterexample problems and fine-tuning the LLMs. Additionally, our work follows the informal-to-formal paradigm for formal proof generation. Different from prior approaches that tightly align informal and formal reasoning (Jiang et al., 2022), we adopt a sequential scheme: the LLM first engages in informal reasoning to search for counterexamples, while formal reasoning is employed to verify the validity of the candidate results through proof.

### 5.3 FORMAL DATA SYNTHESIS

Data plays a crucial role in enhancing the reasoning capability of LLMs (Zeng et al., 2024; Yang et al., 2024; Fleureau et al., 2024). Beyond manual collection and annotation, formal data synthesis methods are typically categorized into two categories: neural and symbolic. Neural methods employ LLMs to either autoformalize existing informal problems or directly generate formal ones. For instance, Minimo (Poesia et al., 2024) and STP (Dong & Ma, 2025) use an LLM conjecturer to propose new theorems, while DeepSeek-Prover (Xin et al., 2024), Kimi-Prover (Wang et al., 2025), and Goedel (Lin et al., 2025a) fine-tune autoformalization models to translate math competition problems into Lean theorems. Symbolic methods often employ a precise and controllable mutation mechanism to ensure the validity of the generated problems. MetaGen (Wang & Deng, 2020) mutates Metamath theorems via substitution, and works such as LeanNavigator (Yin & Gao, 2025) and Alchemy (Wu et al., 2024a) extend this to Lean 4 by applying tactics like *apply* or *rw*. Although broadly applicable, these operators often fail to produce sufficiently hard problems. Thus, some focus on domain-specific strategies, e.g., geometry (Trinh et al., 2024), inequalities (Wei et al., 2024), or TCS (Zhang et al., 2025b), yielding stronger training performance.

Our proposed method falls into the second category; we target the counterexample problem, devise a dedicated mutation strategy, and implement it as a new Lean 4 tactic. This ensures validity while increasing problem diversity and difficulty, and can be applied to neural-generated theorems as well. Based on the mutation strategy, we further design a multi-reward function compatible with popular reinforcement learning methods like GRPO (Shao et al., 2024) and its variants (Zhang et al., 2025a).

## 6 LIMITATIONS.

While our proposed framework demonstrates strong potential in counterexample search and formal proof generation, several directions remain for further improvement and extension.

**Data quality of synthetic data.** Compared with manually collecting and labeling problems and solutions, symbolic mutation can generate large volumes of data at extremely low cost. However, data quality remains an extensive concern, as the generated data often lacks careful selection. This results in the downgrade of training efficiency. For instance, as shown in our training curves (Figure 4), LLM fine-tuning converges within roughly half an epoch, suggesting that a substantial portion of the generated data is redundant. Although prior studies have explored methods to improve synthetic data quality (Muennighoff et al., 2025; Xia et al., 2024; Wang et al., 2024a), effectively incorporating these techniques into formal counterexample generation remains an open challenge.

**Limited budget of model training.** We provide additional case studies in Appendix E, which demonstrate that the limited capability (or size) of the LLM significantly compromises the performance of our framework. First, the 7B informal reasoning model frequently fails to produce correct results, even for simple calculations. Employing a larger model or incorporating tool-use strategies could help alleviate this issue. Second, the 7B formal reasoning model often struggles to follow instructions and fails to utilize the provided counterexamples when deriving formal proofs. This leads to inaccurate feedback rewards that undermine overall performance. While these limitations could likely be mitigated by using larger LLMs, resource constraints prevent us from exploring this direction at present. We leave it for future work and further investigation.

## 7 CONCLUSION

In this work, we investigated the crucial yet largely underexplored problem of formal counterexample generation with reasoning large language models. We identified and discussed two key challenges, data scarcity and sparse reward signals, that hinder the development of models specialized for this task. To address these issues, we introduced an integrated framework that combines a symbolic mutation strategy for large-scale counterexample synthesis with a multi-reward scheme for more effective and efficient model training. Experiments on three newly constructed benchmarks demonstrate that our approach substantially outperforms existing state-of-the-art LLMs in counterexample generation. We believe that our proposed framework opens up a promising direction for enhancing the self-reflective and self-corrective reasoning capabilities of LLMs, while also providing a practical copilot for validating mathematical conjectures.

ETHICS STATEMENT

This work focuses on formal counterexample generation and does not involve human subjects or sensitive data. The datasets employed are publicly available and widely used in prior research. We do not anticipate any ethical concerns or harmful applications arising from this study.

REPRODUCIBILITY STATEMENT

We provide detailed descriptions of the training and inference setup in the main text (Section 4) and Appendix C. All datasets and preprocessing steps are thoroughly documented. Anonymized source code is included to ensure full reproducibility.

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

## A  USAGE OF LLMs IN PAPER WRITING

We used ChatGPT 5 solely for grammar checking and minor language polishing. The LLM was not involved in research ideation, data analysis, or substantive writing. The authors take full responsibility for the content of this paper.

## B  ILLUSTRATIVE EXAMPLES

Figure 5: An instantiation of our counterexample training framework. The original theorem shown is an intermediate subgoal extracted from DSP+'s proof of AIME-II 2001 Problem 3 in miniF2F.

We further illustrate our framework using a concrete example in Figure 5. Starting from the original theorem `aimeII_2001_p3_g4_extracted_54`, the mutation tactic discards $h_5$ and produces two related problems: the counterexample instance `aimeII_2001_p3_mut_54_drop4` and the dropped-hypothesis version `aimeII_2001_p3_54_drop4`. During expert iteration, the LLM first proposes a candidate counterexample (a concrete sequence), and then generates formal proofs for both the mutated and dropped-hypothesis theorems. Once these proofs are verified, the resulting correctness signals are used to compute rewards, which serve as weights for this training example.

## C  EXPERIMENTAL SETTING

In our expert iteration framework, we employ two LLMs; Qwen3 8B (Qwen Team, 2025) for generating candidate counterexamples and Deepseek-Prover-V2 7B (Ren et al., 2025) for producing formal proofs. We fix the maximum generation length to 4,096 tokens and set the sampling temperature to 0.9 for both models. For weighted training, we adapt LLaMA-Factory (Zheng et al., 2024), using its default configuration (Adam optimizer (Kingma, 2014) with a learning rate of 1e-5 and cosine learning rate schedule). Following prior work (Dong & Ma, 2025; Lin et al., 2025b), we perform an additional re-training step on the collected dataset paired with verified correct answers.

During evaluation, for each problem, we prompt Qwen3 to generate three counterexample candidates, and for each candidate, Deepseek-Prover-V2 produces three corresponding formal proofs. For verification, we employ lean-repl with a 60-second timeout and an 8GB memory limit per proof.

Regarding the prompt setting, for the baseline methods, we adopt the official prompt templates provided in their respective repositories. In our method, the prompts are designed for Qwen3 and Deepseek-Prover-V2, and are detailed in the following.

---

**Prompt of Informal Reasoning for Counterexample Search**

Find a concrete example to prove the following existential problem.
Note that:
1. Please reason the problem and give the final answer in Natural Language.
2. The final answer should be in the format \\boxed{{...}}.
The problem is: {formal_statement}

---

> **Prompts of Formal Reasoning for Proof Generation**
>
> Complete the following Lean 4 code using the given concrete example {example}:
>
> ```lean4
> {header}
> {formal_statement}
> ```

## D ADDITIONAL RESULTS

Table 3: Pass@$k$ ($k = 1, 4, 9$) rates on the validation set. The results demonstrate that our framework enables the fine-tuned model to significantly outperform baselines.

| Model | Leanabell | STP | Kimina-distill | Goedel-v2 | Deepseek-v2 | Ours | $\Delta$ |
|---|---|---|---|---|---|---|---|
| ✔@1 | 25.4% | 23.7% | 9.8% | 14.0% | 14.1% | **49.8%** | 24.4%↑ |
| ✔@4 | 36.0% | 33.2% | 25.3% | 31.1% | 30.0% | **52.7%** | 16.7%↑ |
| ✔@9 | 39.9% | 37.5% | 35.8% | 38.2% | 36.2% | **54.1%** | 14.2%↑ |

We also evaluate the performance of our fine-tuned model against existing neural theorem provers on the split-out validation set. As shown in Table 3, the fine-tuned model substantially outperforms the existing approaches by a large margin (14.2% to 24.4%). This improvement can be attributed to the high quality of our synthesized dataset and the effectiveness of multi-reward training.

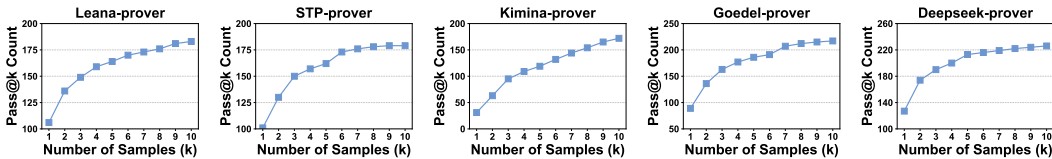

Figure 6: Pass@$k$ curves of five neural theorem provers. The results are derived from the evaluation of FOR-COUNTER and show that the performance nearly converges when $k = 10$.

We plot the pass@$k$ curves in Figure 6 to explain why we select only $k = 1, 4, 9$ in our empirical evaluation. The results show that the performance of the five existing neural theorem provers nearly converges once $k$ approaches 10. Therefore, using $k = 1, 4, 9$ is sufficient to capture the performance differences in our comparison.

Table 4: Pass@$k$ ($k = 1, 4, 9$) rates for different module combinations. Results evaluated on the validation set and FOR-COUNTER show that both modules (informal reasoning and formal reasoning models) are successfully improved by fine-tuning with our framework.

| Model | Pass@$k$ | Qwen3×Ds-prv-v2 | Ours×Ds-prv-v2 | Qwen3×Ours | Ours |
|---|---|---|---|---|---|
| | ✔@1 | 26.2% | 30.9% | 47.2% | 49.8% |
| Validation Set | ✔@4 | 36.2% | 38.7% | 50.3% | 52.7% |
| | ✔@9 | 40.4% | 43.7% | 51.0% | 54.1% |
| | ✔@1 | 14.1% | 15.2% | 19.1% | 20.9% |
| FOR-COUNTER | ✔@4 | 18.9% | 20.1% | 24.3% | 25.8% |
| | ✔@9 | 21.0% | 21.5% | 27.7% | 28.5% |

We conduct an ablation study to evaluate the contribution of each component to the overall performance. Specifically, our method first employs the fine-tuned Qwen3 to propose a candidate counterexample, and then leverages Deepseek-prover-v2 (denoted as Ds-prv-v2) to generate the formal proof based on this counterexample. To examine the necessity and effectiveness of each component, we replace the fine-tuned Qwen3 with its original version, and Ds-prv-v2 with its original version,

respectively. We also include the evaluation results for the combination of the original Qwen3 and the original Ds-prv-v2 as a reference.

The overall results are presented in Table 4. They demonstrate that both modules are necessary, and combining our two fine-tuned models yields the best performance. Compared to the gains from fine-tuning the formal reasoning model, the improvements from fine-tuning the informal reasoning model are relatively marginal. We therefore conduct case studies in the next section to analyze the underlying reasons.

## E   CASE STUDIES

We continue with the running example in Figure 5, which is collected from the validation set, to conduct the case study to demonstrate the typical issue of our current framework. The original theorem, as well as its corresponding proof, is shown below.

```
theorem aimeII_2001_p3_g4_extracted_54
    (x : ℕ → ℤ)
    (h₁ : ∀ n ≥ 5, x n = x (n - 1) - x (n - 2) + x (n - 3) - x (n - 4))
    (h₂ : x 10 = -267)
    (h₃ : x 11 = 211)
    (h₄ : x 12 = 375)
    (h₅ : x 13 ≠ 420) :
    x 14 ≠ 523 := by
  have h6 : x 13 = x 12 - x 11 + x 10 - x 9 := by
    rw [h₁ 13 (by omega)]
  have h7 : x 14 = x 13 - x 12 + x 11 - x 10 := by
    rw [h₁ 14 (by omega)]
  rw [h7]
  rw [h₂, h₃, h₄]
  simp
  intro h
  have : x 13 = 420 := by
    linarith
  exact h₅ this
```

Then, we derive the mutated version by dropping the last hypothesis $h_5$ : x 13 ≠ 420. The counterexample problem is shown as follows.

```
theorem aimeII_2001_p3_mut_54_drop4 :
    ∃ (x : ℕ → ℤ),
    (∀ n ≥ 5, x n = x (n - 1)  - x (n - 2) + x (n - 3)  - x (n - 4))
    ∧ x 10 = -267  ∧ x 11 = 211  ∧ x 12 = 375 ∧ x 14 ≠ 523 := by
  sorry
```

Note that this problem is provable since the dropped hypothesis $h_5$ is necessary for the original theorem. We generate nine responses using our fine-tuned model and find that none of them are fully correct. The main failures can be divided into two cases: (i) incorrect informal reasoning; (ii) incorrect instruction-following. We first show the case (i) as follows.

---

**Prompt of Informal Reasoning for aimeII_2001_p3_mut_54_drop4**

Find a concrete example to prove the following existential problem.
Note that:
1. Please reason the problem and give the final answer in Natural Language.
2. The final answer should be in the format \\boxed{{...}}.
The problem is:
```
theorem aimeII_2001_p3_mut_54_drop4 :
    ∃ (x : ℕ → ℤ),
    (∀ n ≥ 5, x n = x (n - 1)  - x (n - 2) + x (n - 3)  - x (n - 4))
    ∧ x 10 = -267  ∧ x 11 = 211  ∧ x 12 = 375 ∧ x 14 ≠ 523 := by
  sorry
```

---

> **Response of Informal Reasoning Model (Incorrect)**
>
> We are given the recurrence relation
> $$x(n) = x(n-1) - x(n-2) + x(n-3) - x(n-4), \quad n \geq 5$$
> together with the constraints
> $$x(10) = -267, \quad x(11) = 211, \quad x(12) = 375, \quad x(14) \neq 523.$$
>
> **Step 1. Characteristic polynomial.** The recurrence has characteristic equation
> $$r^4 - r^3 + r^2 - r = r(r-1)(r^2+1) = 0,$$
> with roots $0, 1, i, -i$. Thus the general solution is
> $$x(n) = A + Ci^n + D(-i)^n.$$
>
> **Step 2. Apply conditions.** For $n = 10, 11, 12$ we compute:
> $$x(10) = A - C + D = -267,$$
> $$x(11) = A - iC - D = 211,$$
> $$x(12) = A + C + D = 375.$$
>
> From the first and third equations: $2C = 642 \implies C = 321$, and $A + D = 54$. Substituting into the second gives
> $$D = -\tfrac{157+321i}{2}, \qquad A = 54 - D = 54 + \tfrac{157+321i}{2}.$$
>
> **Step 3. Check $x(14)$.** Since $i^{14} = -1$ and $(-i)^{14} = -1$,
> $$x(14) = A - C - D = (A + D) - C - 2D = -267 - 2D.$$
>
> Plugging in $D$ yields
> $$x(14) = -110 + 321i,$$
> which is not a real number and in particular $\neq 523$.
>
> **Conclusion.** A valid function $x$ exists. One explicit example is
> $$x(n) = \left(54 + \tfrac{157+321i}{2}\right) + 321\, i^n - \tfrac{157+321i}{2}\,(-i)^n.$$
>
> This satisfies the recurrence, the given values, and $x(14) \neq 523$.

The response is clearly incorrect, since $x$ should be a real sequence rather than a complex one, which in turn caused the subsequent formal proof generation to fail. In case (ii), the response of informal reasoning is correct, as shown in the next example.

> **Response of Informal Reasoning Model (Correct)**
>
> We can explicitly construct such a sequence by fixing the first four values and then extending by the recurrence. Define
> $$x(1) = 211, \quad x(2) = 375, \quad x(3) = -103, \quad x(4) = 0,$$
> and for all $n \geq 5$ set
> $$x(n) = x(n-1) - x(n-2) + x(n-3) - x(n-4).$$
>
> **Step 1. Compute terms.** Applying the recurrence successively:
> $$x(5) = -211 + 375 - (-103) + 0 = 267, x(6) = -211, x(7) = -375,$$
> $$x(8) = 103, x(9) = 0, x(10) = 211 - 375 + (-103) - 0 = -267,$$
> $$x(11) = 211, x(12) = 375, x(13) = -103, x(14) = 0.$$
>
> **Step 2. Verify conditions.**
> $$x(10) = -267, \quad x(11) = 211, \quad x(12) = 375, \quad x(14) = 0 \neq 523.$$
>
> All required conditions are satisfied.
>
> **Remark.** This sequence is in fact periodic with period 10. In Lean, one can constructively define $x : \mathbb{N} \to \mathbb{Z}$ either by recursion from the initial values, or directly by cases depending on $n \bmod 10$.

Although the informal reasoning is correct and the model provides an accurate counterexample, the formal reasoning model may not adopt it directly. Instead, it may propose an alternative but incorrect one and then generate a formal proof for it. The following example illustrates this typical case.

---

**Prompt of Formal Reasoning for aimeII_2001_p3_mut_54_drop4 (Incorrect)**

Complete the following Lean 4 code using the given concrete example
$x(10) = -267, \quad x(11) = 211, \quad x(12) = 375, \quad x(14) = 0 \neq 523$:

```lean4
theorem aimeII_2001_p3_mut_54_drop4 :
  ∃ (x : ℕ → ℤ),
    (∀ n ≥ 5, x n = x (n - 1)  - x (n - 2) + x (n - 3)  - x (n - 4))
    ∧ x 10 = -267  ∧ x 11 = 211  ∧ x 12 = 375 ∧ x 14 ≠ 523 := by
  /-
  We need to find a function \( x : \mathbb{N} \to \mathbb{Z} \) that satisfies
    the recurrence relation \( x(n) = x(n - 1) - x(n - 2) + x(n - 3) - x(n - 4)
    \) for \( n \geq 5 \), and also satisfies \( x(10) = -267 \), \( x(11) =
    211 \), \( x(12) = 375 \), and \( x(14) \neq 523 \).

  We can define such a function by specifying its values at each integer up to
    \( n = 14 \), ensuring that the recurrence relation is satisfied for \( n
    \geq 5 \). We then verify that the specific values for \( x(10) \) to \(
    x(14) \) meet the given conditions.
  -/
  use fun n => if n = 10 then -267 else if n = 11 then 211 else if n = 12 then
    375 else if n = 13 then 523 else 0
  constructor
  · intro n hn
    simp [Nat.le_of_lt_succ, Nat.le_of_lt_succ] at hn
    split_ifs <;> omega
  constructor
  · simp
  constructor
  · simp
  constructor
  · simp
  simp
  <;> norm_num
  <;> omega
```

