# OpenReview forum: "Learning to Disprove: Formal Counterexample Generation with Large Language Models"
_ICLR.cc/2026/Conference — ICLR 2026 Conference Withdrawn Submission_

### Official Review · Reviewer_v9fU · 2025-10-30

**Soundness:** 2
**Presentation:** 2
**Contribution:** 2
**Rating:** 2
**Confidence:** 4

**Summary:**

This paper proposes a framework for learning to disprove mathematical statements by training large language models (LLMs) to generate and formally verify “counterexample problems” in Lean.
The pipeline begins with valid Lean theorems, automatically mutates them by dropping one hypothesis, and then trains paired models:
(1) an informal reasoner that proposes a natural-language sketch of a potential counterexample, and
(2) a formal reasoner that completes a corresponding Lean proof.
The authors claim this process enables sequential informal-to-formal reasoning for counterexample generation and report improved Pass@k accuracy over baseline models on a custom evaluation set derived from these mutations.

**Strengths:**

- Tackling disproof rather than proof generation is a valuable direction for formal-reasoning LLMs.
- The “hypothesis-dropping” approach provides a practical way to generate large numbers of training problems automatically from existing Lean libraries.
- Splitting the task into informal and formal stages aligns with ongoing efforts to model human-like reasoning pipelines (sketch → formalization).

**Weaknesses:**

* **Fundamental logical mis-specification.**
  The central “counterexample problem” is formulated as an existential statement asserting
  $$
  \exists x,\; (H_1(x)\wedge …\wedge H_{k-1}(x)) \wedge C(x),
  $$
  rather than
  $$
  \exists x,\; (H_1(x)\wedge …\wedge H_{k-1}(x)) \wedge \neg C(x).
  $$
  As a result, the system *proves* an existential claim rather than *disproves* a universal one.
  The paper repeatedly describes these as “counterexample problems,” but they are semantically consistent with the original theorem, not contradictory to it.
  This inversion of logical polarity invalidates much of the “disproof” framing.

* **Misinterpretation of necessity and provability.**
  Phrases such as *“this problem is provable since the dropped hypothesis is necessary”* reverse the correct logic:
  if a hypothesis is necessary, the weakened statement should be *false*, and only its **negation** should be provable.

* **Misalignment with COUNTERMATH.**
  The COUNTERMATH benchmark evaluates *true* counterexamples of the form
  $$
  \exists x,\; P(x)\wedge \neg Q(x),
  $$
  whereas the paper’s tasks remain existential satisfiability problems closer to
  $$
  \exists x,\; P(x)\wedge Q(x).
  $$
  Consequently, any claimed connection or comparison to COUNTERMATH is conceptually incorrect.

* **Evaluation ambiguity.**
  Because the generated statements are not genuine counterexamples, the reported proof success rates do not demonstrate disproof capability.
  The metrics thus measure syntactic Lean proof generation, not semantic reasoning or falsification.

**Questions:**

1. **Clarify the logical formulation.**
   Could the authors explicitly define the intended “counterexample problem” in standard logical form  $ \exists x,\; P(x)\wedge \neg Q(x) $ and explain why the current implementation uses $P(x)\wedge Q(x)$  instead?

2. **Meaning of “provable” counterexample.**
   When the paper says *“the problem is provable since the dropped hypothesis is necessary,”* what is meant—syntactic provability in Lean, or semantic falsity of the original theorem?

---

### Official Review · Reviewer_BEoX · 2025-10-30

**Soundness:** 2
**Presentation:** 3
**Contribution:** 2
**Rating:** 6
**Confidence:** 3

**Summary:**

The paper addresses the neglected challenge of disproving mathematical statements using LLMs. The authors formalize the task as formal counterexample generation, requiring models to both propose a counterexample and produce Lean 4–verifiable proofs. They introduce a symbolic mutation strategy that creates synthetic counterexample data by dropping hypotheses from proven theorems, and a multi-reward expert-iteration framework that uses dual proof signals to stabilize sparse-reward training. Using Qwen3-8B for informal reasoning and DeepSeek-Prover-v2-7B for formal proof generation, their system outperforms prior LLM-based provers across three new benchmarks (FOR-COUNTER, VERI-REASON, VERI-FORMALIZE).

**Strengths:**

* **Originality of the task formalization:** The paper introduces a novel task to disproving mathematical statements using LLMs. The formalization with Lean 4–verifiable proofs is a meaningful.

* **Methodological validity:** The symbolic mutation strategy and multi-reward expert-iteration framework are well-motivated and effectively address the challenges of sparse-reward training.

* **Empirical significance:** The system's performance on three new benchmarks demonstrates its potential to advance the state of the art in formal counterexample generation.

**Weaknesses:**

* **Clarity**: Although I read the paper carefully, I found some parts hard to follow. For instance, the evaluation metrics are not specified, benchmark construction lacks detail and has typos, and hyperparameter settings are not clearly described.

* **Evaluation gaps:** The inference system of the proposed method is two-model but other baselines are single-model. This makes it hard to attribute the performance gain solely to the proposed training method even with the ablation study. More analysis is needed to analyze (1) how different informal&formal model combinations affect the final performance, and (2) how the performance changes if only one model is trained and used in the proposed training framework.

* **Theory/assumptions:** The symbolic mutation strategy assumes that dropping hypotheses from proven theorems will yield valid counterexamples. However, this assumption may not always hold without affirming the dropping hypothesis, and even the "correct" example in Appendix E does not negate "h5". If only the mutated theorems are used for generating the counterexample and for evaluation without noting the dropped hypothesis, it would be problematic.

* **Organization:** The single-reward ablation is superficial since both rewards are apparently necessary. It is better to replace it with other ablations such as different informal&formal model combinations and using single models in the proposed training framework to align with baselines.

Due to the reasons above, I could hardly raise my ratings without substantial revisions from the authors.

**Questions:**

* Can you provide more details on the evaluation metrics used in the experiments? Data points in the test set will be helpful.

* What are the point of VERI-REASON and VERI-FORMALIZE benchmarks? I could grasp some motivation but it is still unclear to me.

* What is the difference of finding the counterexample of a false conjecture and disproving a mathematical statement? Which should be the main focus of the area?

---

### Official Review · Reviewer_fHiZ · 2025-11-01

**Soundness:** 3
**Presentation:** 3
**Contribution:** 3
**Rating:** 6
**Confidence:** 3

**Summary:**

This paper focuses on the understudied task of **formal counterexample generation** in AI mathematical reasoning, a critical complement to proof construction. It proposes a two-stage framework: first, a **symbolic mutation strategy** synthesizes 575K counterexample problems by removing necessary hypotheses from provable theorems, addressing data scarcity. Second, a **multi-reward expert iteration framework** leverages dual rewards (for mutated theorems and dropped hypotheses) to alleviate sparse reward issues. Evaluated on three new benchmarks, the approach achieves 47–74% relative improvements in pass@1 over baselines, with counterexamples verifiable by Lean 4. This work systematically advances counterexample-driven reasoning, aligning with the "positive hard, negative easy" intuition in mathematical problem-solving.

**Strengths:**

- **Novel Problem Focus**: Fills a crucial gap in AI math reasoning by centering formal counterexample generation, a practice rooted in the "positive hard, negative easy" heuristic (e.g., using counterexamples when proofs stall).
- **Innovative Data Synthesis**: The symbolic mutation strategy creatively generates large-scale, high-quality counterexample data (575K problems) from diverse theorem sources, solving the data scarcity bottleneck.

**Weaknesses:**

- **Informal-Formal Consistency Gaps**: The pipeline risks inconsistencies between informal natural language counterexamples and formal Lean 4 proofs—e.g., ambiguous natural language descriptions, implicit assumptions, or misaligned predicate definitions may fail formal verification.
- **Synthesized Data Redundancy**: The paper notes redundant synthetic data causes early convergence (half an epoch) but lacks detailed analysis of *why* redundancy occurs (e.g., duplicate problem structures, overlapping logical patterns) or its impact on generalization.
- **Human-Model Counterexample Comparison**: There is no evaluation of how model-generated counterexamples compare to human-annotated ones in terms of quality, creativity, or logical rigor, leaving uncertainty about real-world utility.

**Questions:**

1. Regarding informal-formal consistency: Does the paper encounter failures where natural language counterexamples fail formal verification? If so, what are the main causes (e.g., semantic ambiguity, predicate mismatches), and are there mitigation strategies (e.g., intermediate formalization checks, constrained prompt engineering)?
2. On synthetic data redundancy: Could you elaborate on the nature of redundancy (e.g., duplicate problems, similar logical structures) and why it leads to early convergence? Are there data pruning or diversification strategies to address this?
3. For human-model comparison: Have you evaluated model-generated counterexamples against human-annotated ones (e.g., in terms of correctness, elegance, or alignment with mathematical intuition)? If not, could this be included in future work?
4. Generalizability to Existing Benchmarks: The paper evaluates performance solely on newly proposed benchmarks. Can the model also achieve improvements on established mathematical reasoning benchmarks (e.g., CounterMath, MiniF2F) that involve counterexample reasoning? This would help assess its broader utility beyond the custom benchmarks.

**Details Of Ethics Concerns:**

**Privacy and Anonymity Breach Risk**: The supplementary material contains a Hugging Face login token, which could potentially reveal the authors’ identity, maybe violating the double-blind review protocol of ICLR.

---

### Official Review · Reviewer_pPw7 · 2025-11-02

**Soundness:** 3
**Presentation:** 3
**Contribution:** 2
**Rating:** 6
**Confidence:** 2

**Summary:**

The paper tackles formal counterexample generation for mathematical theorems using LLMs. The key contributions are a symbolic mutation strategy that creates counterexample problems by dropping necessary hypotheses from proven theorems, and a multi-reward training framework that provides intermediate rewards even when complete counterexamples aren't found. They synthesize 575K training problems and achieve 47-74% improvements over baselines on three new benchmarks.

**Strengths:**

- Addresses an important, underexplored problem in mathematical reasoning
- The hypothesis-dropping mutation strategy is principled and guarantees valid problems while maintaining reasonable diversity
- Multi-reward framework elegantly handles the sparse reward problem that would otherwise limit training to easy problems
- Strong empirical results with 47-74% relative improvements across multiple benchmarks

**Weaknesses:**

- Lacks theoretical analysis of what makes the generated problems effective for training - why does this particular distribution of problems work?
- Scalability concerns: mutation generates only ~1.78x problems per seed; training saturates after half an epoch suggesting significant redundancy; approach requires existing formal libraries
- Model size limitations fundamentally constrain performance - case studies show 7-8B models make basic arithmetic errors and fail to follow instructions, but resource constraints prevented testing larger models
- Data quality vs quantity tradeoff not addressed - generates 575K problems but acknowledges redundancy without exploring filtering or selection strategies
- Insufficient experimental details: single epoch training seems premature; α parameter choice appears arbitrary; validation set selection process unclear

**Questions:**

- What analysis have you done on the properties of effective counterexample problems? Is there a relationship between problem complexity/structure and training value?
- Why not implement quality filtering or curriculum learning given the acknowledged data redundancy? Have you analyzed which problems contribute most to performance gains?
- How does the computational cost compare to symbolic methods for problems both can handle? What's the practical tradeoff?
- The mutation ratio varies significantly (1.24-2.48) across datasets - what drives this variation and does it correlate with training effectiveness?
- Can you provide more thorough ablation on the α parameter? The single comparison (0.8 vs 1.0) isn't sufficient to understand this design choice.

---

### Official Review · Reviewer_A4FQ · 2025-11-05

**Soundness:** 1
**Presentation:** 3
**Contribution:** 1
**Rating:** 2
**Confidence:** 5

**Summary:**

The paper studies formal counterexample generation: given a universally-quantified Lean 4 theorem, they drop a hypothesis to synthesize a new task and train LLMs to propose a counterexample in NL and produce a Lean-verifiable proof that the counterexample works. The core is a symbolic mutation tactic (drop a hypothesis from a provable theorem) plus a multi-reward expert-iteration loop that rewards proving both the mutated goal and the “dropped-hypothesis” goal.

**Strengths:**

1. The formalization of counterexample generation as an informal-to-formal task in Lean 4 is a valuable direction for the community.
2. The Symbolic Mutation Strategy does successfully create a large volume (575K) of synthetically generated problems, which is a practical contribution.

**Weaknesses:**

1. The unacceptably large standard deviation relative to the mean reported in the results (e.g., training curves and ablations) makes the empirical claims of superiority unreliable and unsubstantiated. This severe instability suggests the method is not robust and the reported mean performance is likely a statistical artifact of favorable seed initialization. Without robust, repeatable results, the paper cannot be accepted.

2. Both core techniques, Multi-Reward RL and Symbolic Mutation, lack fundamental novelty. The authors fail to adequately contextualize these techniques against prior work (e.g., mutation testing in formal methods or auxiliary reward schemes in RL), making the claims of originality misleading.

3. Using an insufficient 7B base model leaves a major confounding variable: is the low performance ceiling due to the base model's limitations or the framework's failure? The inability to effectively pass the correct counterexample from the informal stage to the formal stage further highlights this execution failure.

**Questions:**

The high variance is a critical flaw. Can you provide a rigorous analysis identifying the source of the instability (e.g., policy updates, environment noise, or hyperparameter sensitivity)? More critically, you must re-run and demonstrate that the STD has been dramatically reduced (e.g., at least halved) while maintaining the reported mean performance, or the empirical results must be discarded as non-significant?

Since the core ideas are not fundamentally new, please provide citations and a discussion distinguishing Symbolic Mutation from mutation testing (e.g., in MuZOT or other theorem-proving test generators) and Multi-Reward from established auxiliary reward/curiosity methods in RL. Specifically, what makes your application of these established concepts novel beyond domain-specific tailoring?

Given the known weakness of the 7B base model, what is the zero-shot performance of a much stronger, modern general-purpose LLM (e.g., a 70B parameter model) on the informal counterexample generation task? Without this, it is impossible to gauge the true value added by the complex RL fine-tuning framework.

---

### Note · Authors · 2025-11-21

**Comment:**

Due to limited research resources, we are unable to complete the necessary updates and experiments within the required timeframe. Therefore, we have decided to withdraw this submission. We sincerely appreciate the reviewers and area chairs for their time and constructive feedback.

**Withdrawal Confirmation:**

I have read and agree with the venue's withdrawal policy on behalf of myself and my co-authors.